# Antiviral Immunoglobulins of Chicken Egg Yolk for Potential Prevention of SARS-CoV-2 Infection

**DOI:** 10.3390/v14102121

**Published:** 2022-09-26

**Authors:** Erlend Ravlo, Lasse Evensen, Gorm Sanson, Siri Hildonen, Aleksandr Ianevski, Per Olav Skjervold, Ping Ji, Wei Wang, Mari Kaarbø, Gerda Dominyka Kaynova, Denis E. Kainov, Magnar Bjørås

**Affiliations:** 1Department of Clinical and Molecular Medicine (IKOM), Norwegian University of Science and Technology, 7028 Trondheim, Norway; 2Norimun AS, Felleskjøpet Agri SA, Postboks 469, 0105 Oslo, Norway; 3Felleskjøpet Fôrutvikling AS, Nedre Ila 20, 7018 Trondheim, Norway; 4Department of Microbiology, Oslo University Hospital, 0105 Oslo, Norway; 5Thora Storm School, 7012 Trondheim, Norway; 6Institute of Technology, University of Tartu, 50411 Tartu, Estonia; 7Institute for Molecular Medicine Finland, University of Helsinki, 00014 Helsinki, Finland

**Keywords:** IgY, SARS-CoV-2, Vero-E6, Syrian Golden hamster, antiviral strategy

## Abstract

*Background:* Some viruses cause outbreaks, which require immediate attention. Neutralizing antibodies could be developed for viral outbreak management. However, the development of monoclonal antibodies is often long, laborious, and unprofitable. Here, we report the development of chicken polyclonal neutralizing antibodies against SARS-CoV-2 infection. *Methods:* Layers were immunized twice with 14-day intervals using the purified receptor-binding domain (RBD) of the S protein of SARS-CoV-2/Wuhan or SARS-CoV-2/Omicron. Eggs were harvested 14 days after the second immunization. Polyclonal IgY antibodies were extracted. Binding of anti-RBD IgYs was analyzed by immunoblot and indirect ELISA. Furthermore, the neutralization capacity of anti-RBD IgYs was measured in Vero-E6 cells infected with SARS-CoV-2-mCherry/Wuhan and SARS-CoV-2/Omicron using fluorescence and/or cell viability assays. In addition, the effect of IgYs on the expression of SARS-CoV-2 and host cytokine genes in the lungs of Syrian Golden hamsters was examined using qRT-PCR. *Results:* Anti-RBD IgYs efficiently bound viral RBDs in situ, neutralized the virus variants in vitro, and lowered viral RNA amplification, with minimal alteration of virus-mediated immune gene expression in vivo. *Conclusions:* Altogether, our results indicate that chicken polyclonal IgYs can be attractive targets for further pre-clinical and clinical development for the rapid management of outbreaks of emerging and re-emerging viruses.

## 1. Introduction

Viral diseases consistently pose a substantial economic and public health burden. This burden is due to the ability of viruses to be transmitted from wild and domestic animals to humans, resulting in unpredictable outbreaks. The current strategy for viral outbreak management is heavily reliant on broad-spectrum antivirals and vaccines [1]. While the antiviral drug repurposing and development of novel vaccines are often long, laborious, and unprofitable, neutralizing antibodies remain an effective disease management option [2].

Several recombinant monoclonal antibodies that bind to the viral trimeric S protein and prevent binding with the host receptor angiotensin-converting enzyme 2 (ACE2) have been approved or have entered clinical trials. They have shown a preliminary clinical impact in limiting the development of severe COVID-19 when administered during the early onset of infection [3,4,5]. However, intravenous administration, high concentrations, and high costs represent important limitations to the accessibility of these therapeutics. Thus, the need for additional, more efficacious approaches remains. 

Polyclonal antibodies are easier to produce at low cost. However, many natural sources for the production of polyclonal antibodies have not been fully explored or understood [6]. Chicken egg yolk immunoglobulins (IgYs) possess therapeutic potential, with many advantages [7]. They can readily be generated in large quantities with minimal environmental impact by using egg-laying hens. IgYs showed neutralizing activity in in vitro and in vivo experiments against SARS-CoV, influenza virus, Ebola virus, Zika virus, Dengue virus, human norovirus, and SARS-CoV-2, and had favorable safety profiles [8]. IgYs are fast-acting. IgYs are homologues of mammalian IgGs. However, they can neither bind to Fc receptors nor activate complement components in human; therefore, the exacerbation of viral diseases through antibody-dependent enhancement could be potentially avoided.

Here, we report the development of chicken IgYs against the receptor-binding domain (RBD) of the Spike glycoprotein (S) of the SARS-CoV-2 virus. We show that anti-RBD IgY preparations efficiently bind the antigens in situ, neutralize SARS-CoV-2/Wuhan and SARS-CoV-2/Omicron in vitro, and significantly lower viral replication, with a minimal effect on immune-related gene expression in vivo. The IgY-based approach could have powerful prophylactic potential, which can be leveraged for use in response to other viruses, and their immune-evading or drug-resistant variants. The IgY-based therapeutics could fill the gap between virus identification and drug repurposing and monoclonal antibody and vaccine development.

## 2. Materials and Methods

### 2.1. Preparation and Quantification of Anti-RBD IgY

RBD antigens of SARS-CoV-2 variants (His-tagged RBDs of S proteins of SARS-CoV-2/Wuhan (319–541 aa), the Native Antigen Company, REC31882, and SARS-CoV-2/Omicron (319–552 aa), Sino Biological, 40592-V08H121) were diluted in PBS to obtain 20 µL volume and mixed and emulsified with 180 µL Imject Freunds Complete Adjuvant (Thermo Scientific, Waltham, MA, USA) Each layer (Lohmann LSL Classic) was immunized twice, 14 days apart, via intramuscular injections (breast muscle), with a total volume of 200 µL of RBDs (10 µg) per layer per injection. Then, 14 days after the second immunization, eggs were collected. As a negative IgY control, eggs were collected before the layers were immunized (non-immune IgY). IgYs were extracted from the yolk using the water dilution method. Yolk was separated from the egg white, and 7 volumes of distilled water were added. pH was adjusted to 5 using 1 N HCl. The water-diluted yolk was frozen at −20 °C. The thawed solution was filtered through Whatman Grade 52 Filter Paper for lipid extraction. Then, 8.8% NaCl was added to the filtrate. To precipitate IgY, the pH was adjusted to 4 using 1 N HCl. The solution was stirred gently for 2 h before centrifugation at 3800× *g* for 20 min. The pellets, consisting of the crude extracts of precipitated IgY antibodies, were solubilized in PBS (pH 7).

### 2.2. SDS-PAGE and Immuobloting

The RBDs (0.3 μg) and IgYs (0.5 μg) were separated using SDS-PAGE and stained with Coomassie. For immunoblot analysis, 0.3 μg RBD was separated using SDS-PAGE and transferred to a nitrocellulose membrane (Sartorius). The membranes were blocked with 10% BSA (Santa Cruz Biotechnology) in TBS. Primary IgY antibodies were diluted in TBS (5 μg/mL) and added to the membranes. After 2 h incubation at room temperature, membranes were washed three times with TBS buffer containing 0.03% Tween 20 (Tween/TBS). Secondary antibody conjugated to HRP (1:5000, goat-anti-chicken IgY, GtxCk-003-DHRPX, ImmunoReagents) was added. After 1 h, chemiluminescence was detected using the Western Lightning Chemiluminescence Reagent (Perkin Elmer) by the ChemiDoc Imaging System (BioRad).

### 2.3. Indirect ELISA Assay

The wells of ELISA plates (100 µL/well) were coated overnight at 4 °C with the HEK-293 expressed RBD antigens at a concentration of 20 ng/well (sealing the plates with a film, 391-0624, VWR). The next day, the ELISA plates were emptied in a sink by turning the plates upside down in one vigorous movement. To remove excess liquid, the plates were briefly placed upside down on a paper towel. Next, the plates were washed using an ELISA washer as follows: each well on the ELISA plate was, in a cycle repeated three times, filled and aspirated with 250 µL PBS containing 0.1% Tween. Then, the plates were blocked for 1 h at room temperature by pipetting 100 µL into each well with PBS containing 1% BSA. Meanwhile, the protein concentration of the IgY samples was measured using a Bradford assay. IgY samples with the concentration of 300 µg/mL were prepared. After coating, the ELISA plates were washed as described. Then, all the wells in rows B–H were filled with 100 µL PBS–0.1% Tween and 150 µL of the IgY samples with a concentration of 300 µg/mL were pipetted into row A. All IgY samples were tested in duplicate. To obtain dilution series with a 3x dilution for each step, 50 µL was transferred from row A to row B using a multichannel pipette. This was repeated for all the rows (row B–H) to complete the dilution series (each well now containing 100 µL of the IgY sample). Plates were incubated at room temperature for 1 h. Then, the plates were emptied and washed as described. The HRP-conjugated secondary antibody (goat-anti-chicken-IgY (#A16054, ThermoFisher, Waltham, MA, USA)) was diluted 1:20,000 in PBS–0.1% Tween and 100 µL of this solution was pipetted into each well and incubated for 1 h at room temperature. Then, the plates were emptied and washed as described. TMB solution (Pierce™ TMB Substrate Kit, #34021, ThermoFisher, Waltham, MA, USA) kept at room temperature was prepared and used as recommended by the manufacturer to develop the ELISA assay. Absorption was measured at 650 nm using a plate reader. We performed the experiment twice with 2 parallels.

### 2.4. Anti-RBD IgY Testing and Data Quantification

The engineering of recombinant mCherry-expressing SARS-CoV-2/Wuhan (SARS-CoV-2-mCherry) and propagation of SARS-CoV-2-mCherry and SARS-CoV-2/Omicron have been described previously [9,10,11,12,13,14]. To quantitate the production of infectious virions, the viruses were tittered using plaque assays [9,10,11,12,13]. 

Approximately 4 × 10^4^ Vero-E6 cells were seeded per well in 96-well plates. The cells were grown for 24 h in DMEM supplemented with 10% FBS and Pen–Strep. The medium was then replaced with DMEM containing 0.2% BSA, Pen–Strep, and the IgY preparation in 3-fold dilutions at 7 different concentrations. No IgY preparation was added to the control wells. After 15 min, the cells were infected with virus at an moi of 0.01 or mock. After 48 h of infection, fluorescence of SARS-CoV-2-mCherry-infected cells was measured using the FluoStar Omega plate reader (BMG Labtech). After 48 h of infection, CellTiter-Glo (CTG, Promega, Madison, WI, USA) assays were performed to measure the viability of SARS-CoV-2-mCherry/Wuhan-, SARS-CoV-2/Omicron-, and mock-infected cells, as described previously [15,16,17]. We performed the experiments several times with 3 parallels using several IgY preparations.

The half-maximal effective concentrations (EC_50_) were calculated using the drugvirus.info server [18], based on the analysis of the viability of infected cells, by fitting drug dose–response curves using a four-parameter (4*PL*) logistic function *f (x)*:(1)$$f\left(x\right)=A_{min}+\frac{A_{max}-A_{min}}{1+{\left(\frac{x}{m}\right)}^{\lambda }}$$
where *f (x)* is a response value at dose *x*, *A_min_* and *A_max_* are the upper and lower asymptotes (minimal and maximal drug effects), *m* is the dose that produces the half-maximal effect (EC_50_ or CC_50_), and *λ* is the steepness (slope) of the curve. The relative effectiveness of the drug was defined as the selectivity index (*SI* = CC_50_/EC_50_).

To quantify each drug response in a single metric, a drug sensitivity score (*DSS*) was calculated as a normalized version of the standard area under dose–response curve (*AUC*), with the baseline noise subtracted, and the normalized maximal response at the highest concentration (often corresponding to off-target toxicity):(2)$$DSS=\frac{AUC-t\left(x_{max}-x_{min}\right)}{\left(100-t\right)\left(x_{max}-x_{min}\right)\log_{10}Amin}$$
where activity threshold *t* equals 10%, and DSS is in the 0–50 range [19,20].

### 2.5. Prophylactic Study of Anti-RBD IgY against SARS-CoV-2 Infection in Hamsters

The in vivo hamster study was performed at the OncoDesign Biotechnology facilities in Villebon-sur-Yvette, France. Eighteen Syrian Golden hamsters (6–8-week-old males) were randomly distributed to three groups. Group A was treated with IgY raised against the RBD of the S protein of SARS-CoV-2/Wuhan (4.6 mg IgY/dose) intranasally 1 h before infection and then twice per day for 3 days. The Slovakia/SK-BMC5/2020 virus (B.1.1 lineage, 10^5^ pfu TCID_50_ per animal) was introduced intranasally at Day 0. Group B was infected with the virus (10^5^ pfu TCID_50_ per animal) at Day 0 but IgY was not administered. Group C remained uninfected and untreated. Animal viability, behavior, and clinical parameters were monitored daily. After 4 days, animals were deeply anesthetized using a cocktail of 30 mg/kg (0.6 mL/kg) Zoletil and 10 mg/kg (0.5 mL/kg) Xylazine IP. Cervical dislocation followed by thoracotomy was performed. The superior, middle, post-caval, and inferior lobes of lungs were collected and stored in RNAlater tissue storage reagent overnight at 4 °C, and then at −80 °C.

### 2.6. Gene Expression Analysis

Total RNA was isolated using the RNeasy Plus Mini kit (Qiagen, Hilden, Germany) from the lungs of Syrian hamsters. RT-PCR was performed using the SuperScript™ III One-Step qRT-PCR System kit (1732-020, Life Technologies) with primers ORF1ab_Fw: CCGCAAGGTTCTTCTTCGTAAG, ORF1ab_Rv: TGCTATGTTTAGTGTTCCAGTTTTC, ORF1ab_probe: Hex-AAGGATCAGTGCCAAGCTCGTCGCC-BHQ-1 targeting a region on ORF1ab. Cytokine gene profiling (TNFα, IFNγ, IL-2, IL-4, IL-6, IL-10, IL-12p40, IL-21) was performed using qRT-PCR, as described in the Appendix A [9,21]. RT-qPCR was performed using a Bio-Rad CFX384™ and adjoining software. The relative gene expression differences were calculated using b-Actin as a control, and the results were represented as relative units (RU). Technical triplicates of each sample were performed on the same qPCR plate, and non-templates and non-reverse transcriptase samples were analyzed as negative controls. Statistical significance (*p* < 0.05) of the quantitation results was evaluated with *t*-test. The Benjamini–Hochberg method was used to adjust the *p*-values.

### 2.7. Virus TCID_50_ Determination

The tissue culture infective dose that causes 50% cytotoxicity (TCID_50_) assay is a quantitative method for assessing the infectivity of a virus stock. One TCID_50_ is defined as the amount of pathogen that causes the death of 50% of cells, so TCID_50_ depends on the ability of the virus to kill the cells in culture. Infectivity is expressed as TCID_50_/mL/48 h based on the Spearman–Karber formula. Vero-E6-TMPRSS2 cells were counted, and their viability was assessed by 0.25% trypan blue exclusion assay using the ViCell apparatus. One day before testing, cells were plated in a 96-well plate at the density of 2 × 10^4^ cells per well in a volume of 200 µL of complete growth medium (DMEM, 10% FCS). Cells were infected with serial dilutions of the lung homogenate (triplicate) for 1 h at 37 °C. Fresh medium was added. Then, 48 h after cell infection, an MTS-PMS assay was performed. The assay was performed according to the provider’s protocol (G5430, Promega). After discarding all supernatant, 100 µL of fresh medium and 20 µL of MTS-PMS reagent were added to the culture wells. After a maximum of 4 h, plates were read using an ELISA plate reader and data were recorded (OD value in negative cell control >1.500).

### 2.8. Lung Histology

Left lung specimens harvested at euthanasia were embedded in paraffin (1 slide per animal). Sections of 5 µm thickness were cut and mounted on SuperFrost plus glass slides and stained with H&P (Hematoxylin–Phloxin) to visualize histomorphometry changes. Slides were scanned using the NanoZoomer Digital Pathology System C9600-02 and analyzed using Definiens software. For each section, several criteria were evaluated: (1) recruitment of inflammatory cells for bronchial and alveolar walls; (2) presence of pulmonary edema; (3) presence of alveolar hemorrhage. A scoring grid was used as follows: score 0—no pathological changes observed, edema—absent, hemorrhage—absent; score 1—<10% area affected by pathological changes, edema—present, hemorrhage—present; score 2—10–50% area affected by pathological changes; and score 3—>50% area affected by pathological changes.

## 3. Results

We immunized layers twice, at 14-day intervals, using the purified RBD of SARS-CoV-2/Wuhan or SARS-CoV-2/Omicron per layer per immunization (Figure 1a,b). Eggs were harvested 14 days after the second immunization and IgYs were extracted from yolks (Figure 1b). IgY binding was analyzed by immunoblotting and indirect ELISA (Figure 1c,d). IgYs specifically recognized both Wuhan and Omicron RBDs. IgY from non-immunized layers did not bind RBD. These results demonstrate that our immunization protocol led to the production of RBD-specific IgYs.

We next examined whether anti-RBD IgYs can inhibit SARS-CoV-2 infection and protect cells from virus-mediated death. We tested the antiviral efficacy and toxicity of anti-RBD and control IgYs by monitoring SARS-CoV-2-mediated mCherry expression and the viability of Vero-E6 cells, as described previously [4]. We observed that anti-RBD but not control IgYs reduced SARS-CoV-2-mediated mCherry expression and rescued cells from virus-mediated death (Figure 2a,b). According to both assays, the EC_50_ values were in the 0.1–0.3 mg/mL range and SI were >10, indicating the high selectivity of anti-RBD IgYs. Importantly, IgYs raised against the RBDs of the S proteins of Wuhan and Omicron variants were also able to neutralize SARS-CoV-2/Omicron (Figure 2c).

Next, we examined whether anti-RBD IgY can affect the replication of SARS-CoV-2 in vivo. Eighteen 6–8-week-old male Syrian Golden hamsters were randomly distributed to three groups. Group A was inoculated with SARS-CoV-2 and treated intranasally with anti-RBD IgY 1 h before inoculation and then twice per day for 3 days. Group B was infected with the virus (10^5^ pfu TCID_50_ per animal) at Day 0 but IgY was not administered. Group C remained uninfected and untreated. Animal viability, behavior, and clinical parameters were monitored daily for 4 days. However, no significant difference between group A and B was detected (Figure 3a; Appendix A). 

After 4 days, animals were deeply anesthetized, the cervical region was dislocated, and thoracotomy was performed. The lungs were collected. We did not detect significant differences in lung histology between group A and B (Figure 3b). We were also unable to demonstrate viral neutralization by IgYs in the lungs (Appendix A).

We also extracted RNA from the superior lobes of the right lungs and used RT-qPCR to analyze the expression levels of the ORF1ab gene of SARS-CoV-2 in the lungs at Day 4 after infection. The values were normalized to the expression of the β-Actin gene. We found that anti-RBD IgY significantly reduced the relative levels of ORF1ab RNA in the lungs of infected animals (Figure 3c). We also profiled the expression of several immune-related genes in the lungs of the three groups of animals. We observed a change in the relative RNA levels of IL-2 (*p* < 0.05) between group A and B (Figure 3d). This suggests that treatment with anti-RBD IgY has little effect on the virus-mediated expression of immune-related genes in the lungs of Syrian Golden hamsters.

## 4. Discussion

Here, we report the production of chicken IgYs raised against the RBDs of the S proteins of SARS-CoV-2 variants. We demonstrate the binding and neutralization capacity of the anti-RBD IgYs in situ and in vitro. Importantly, the anti-SARS-CoV-2/Wuhan and anti-SARS-CoV-2/Omicron RBD S IgYs were cross-reactive with the RBDs of the S proteins of SARS-CoV-2/Omicron and SARS-CoV-2/Wuhan, respectively. This result agrees with other studies showing the cross-reactivity and neutralization capacity of IgYs against Alpha (B.1.1.7), Beta (B.1.351), Delta (B.1.617.2), and Omicron (B.1.1.529) [22]. This is different from the neutralization capacity of sera or monoclonal antibodies of immunized or convalescent patients. 

Anti-RBD IgYs lowered viral RNA amplification in the lungs of hamsters challenged with SARS-CoV-2. At the same time, the treatment allowed the activation of several innate immune-related genes that mediate the development of adaptive immune responses. Nevertheless, we were unable to demonstrate viral neutralization (Appendix A). This may be associated with the high virus load (10^5^ TCID_50_ in our study vs. 1 × 10^4^ or 5 × 10^4^ TCID_50_ in published studies [22,23]. In addition, the virus that was delivered in 50 μL of liquid directly into each nare may have washed out some IgYs. Moreover, the IgYs might bind and neutralize some virus particles in the nasal cavity, but remaining virus could still reach the lungs and replicate. 

Only 2–10% of polyclonal chicken IgYs are specific to their antigens [24,25,26,27]. By contrast to purified monoclonal antibodies, such as 23G7 (EC50 in ng/mL range) [9], 100-fold higher concentration of IgYs (EC50 in mg/mL range) is required to achieve an anti-SARS-CoV-2 effect in vitro (Appendix A). The same could be applicable for the in vivo activity of IgYs. Therefore, the purification of antigen-specific IgYs could be an important step to improve the antiviral efficacy of IgYs in vivo.

A recent study demonstrated a protective effect of IN administration of anti-RBD IgY in hamsters challenged with SARS-CoV-2 [24], which suggests that our prophylactic treatment with IgYs, if optimized, could also demonstrate a protective effect. In addition, an anti-SARS-CoV-2-RBD IgY preparation had an excellent safety profile in rats following IN delivery [22]. Moreover, IgYs did not bind/cross-react with a variety of human tissues: these antibodies did not bind the Fc receptor or rheumatoid factor or activate the human complement cascade, thus greatly reducing the risk of severe immune responses [28]. In addition, a double-blind, randomized, placebo-controlled phase 1 study evaluating single-ascending and multiple doses of anti-SARS-CoV-2 RBD IgYs administered IN in healthy adults also demonstrated an excellent safety and tolerability profile, and no evidence of systemic absorption [22]. 

Further studies will be required to demonstrate if the IgY-based approach, which was evaluated by in situ, in vitro, and in vivo models and a phase 1 clinical trial [22,24,26,29,30] could be translated into phase 2 trials. If successful, IgY-based antiviral therapeutics could be valuable due to their lowered potential for adverse side effects (except for patients with egg allergies). In addition, the IgY could be delivered through inhalation and other routes, leading to greater ease of treatment. 

Thus, our and other studies show that research on potential antiviral IgYs can have a significant local as well as global impact, by increasing the protection of the population against emerging and re-emerging viral diseases and filling the gap between virus identification and vaccine/monoclonal antibody/small molecule development [27,31].

## 5. Conclusions

Our pilot study demonstrated the anti-SARS-CoV-2 activity of chicken IgYs. It illustrates an important approach to the development of fast, inexpensive, and effective treatments against SARS-CoV-2. We believe that the further development of antiviral IgY technology could lead to practical therapeutic options against many viruses.

## Figures and Tables

**Figure 1 viruses-14-02121-f001:**
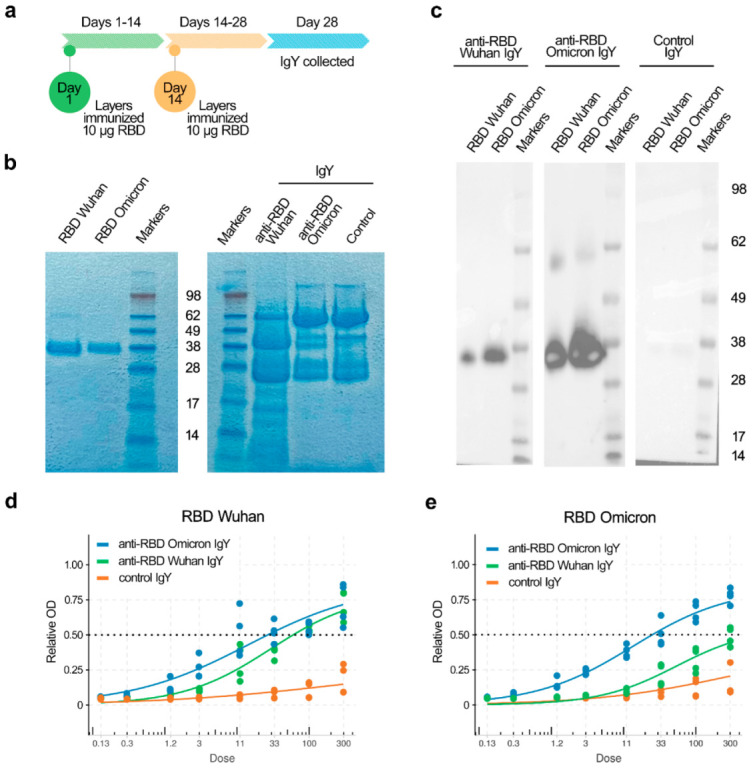
IgY antibodies produced in chicken eggs in response to immunization of layers with receptor-binding domains (RBD) of spike glycoproteins (S) of Wuhan and Omicron SARS-CoV-2 virus variants bind and cross-react with purified RBDs of S proteins of Wuhan and Omicron SARS-CoV-2 variants. (**a**) Layers were immunized twice at 14-day intervals using purified RBD of SARS-CoV-2. IgYs were extracted from eggs harvested 14 days after the second immunization. (**b**) SDS-PAGE of purified RBDs of S of Wuhan and Omicron variants, as well as corresponding anti-RBD and control IgYs. (**c**) Binding of anti-RBD_Wuhan, anti-RBD_Omicron, or control IgY to Wuhan and Omicron RBDs analyzed by immunoblot. (**d**) Binding of anti-RBD_Wuhan, anti-RBD_Omicron, or control IgYs to Wuhan RBD measured by indirect ELISA (OD at 650 nm). Mean; *n* = 4 (2 experiments with 2 parallels). (**e**) Binding of anti-RBD_Wuhan, anti-RBD_Omicron, or control IgYs to Omicron RBD measured by indirect ELISA (OD—650 nm). Mean; *n* = 4 (2 experiments with 2 parallels).

**Figure 2 viruses-14-02121-f002:**
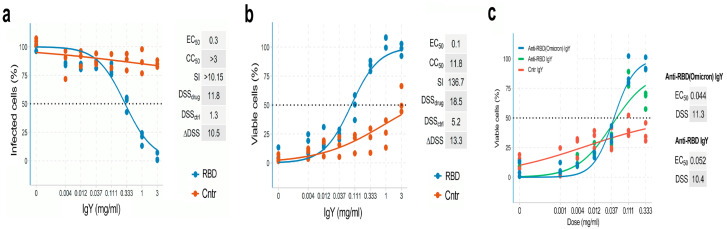
Anti-RBD IgY antibodies produced in chicken eggs neutralize SARS-CoV-2/Wuhan and SARS-CoV-2/Omicron in vitro. (**a**) Neutralization capacity of anti-RBD Wuhan and control non-immune IgYs was measured by monitoring mCherry fluorescence of Vero-E6 cells after 48 h of infection with mCherry-encoded SARS-CoV-2/Wuhan. Mean; *n* = 3. (**b**) Neutralization capacity of anti-RBD Wuhan and control non-immune IgYs was measured by monitoring viability of Vero-E6 cells after 48 h of infection with mCherry-encoded SARS-CoV-2/Wuhan. Mean; *n* = 3. (**c**) Neutralization capacity of anti-RBD Wuhan, anti-RBD Omicron, and control non-immune IgY was measured by monitoring viability of Vero-E6 cells after 48 h of infection with SARS-CoV-2/Omicron. Mean; *n* = 3.

**Figure 3 viruses-14-02121-f003:**
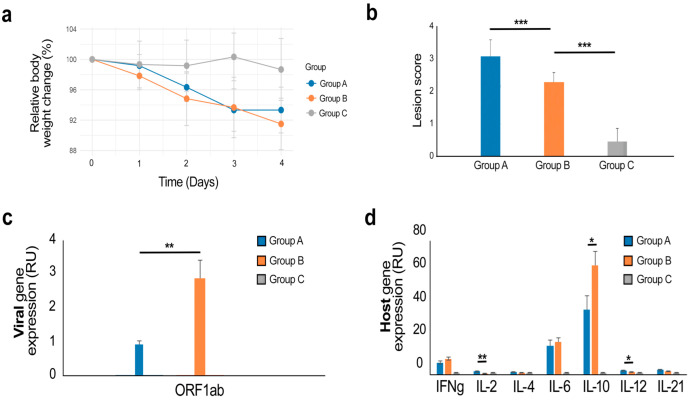
IgY antibodies produced in chicken eggs against RBD of S protein of SARS-CoV-2/Wuhan reduce viral RNA amplification, with minimal alteration of virus-mediated immune gene expression in lungs of infected Syrian Golden hamsters. (**a**) Body weight changes expressed in % in relation to D0 of group A (IgY-treated, virus-infected), group B (virus-infected), and group C (mock-infected) animals (*n* = 6/group). (**b**) Lesion scores in the lungs of Syrian Golden hamsters. Statistical evaluation was performed using Wilcoxon rank-sum test (***—*p* < 0.001). Mean ± SD; *n* = 6. (**c**) Effect of anti-RBD_S_SARS-CoV-2/Wuhan IgYs on the expression of SARS-CoV-2 gene in lungs of Syrian Golden hamsters. Mean ± SD; *n* = 6. *—*p* < 0.1, **—*p* < 0.05 (Wilcoxon test). (**d**) Effect of anti-RBD IgYs on the expression of host cytokine genes in lungs of Syrian Golden hamsters. Mean ± SD; *n* = 6. *—*p* < 0.1, **—*p* < 0.05 (Wilcoxon test).

## Data Availability

All data generated or analyzed during this study are included in this published article and its Appendix A.

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
