# Peer review of "Antiviral Immunoglobulins of Chicken Egg Yolk for Potential Prevention of SARS-CoV-2 Infection"

_viruses, 2022, doi:10.3390/v14102121_

Round 1

Reviewer 1 Report (Previous Reviewer 2)

I have no further comments

Author Response

Thank you!

Reviewer 2 Report (New Reviewer)

Author Response

The manuscript “Antiviral immunoglobulins of chicken egg yolk for prevention of SARS-CoV-2 infection” by E Ravlo et al proposes a strategy to use chicken egg yolk immunoglobulins to control emerging viral infections, using SARS-CoV-2/Covid-19 as a highly relevant example.

The manuscript, while short, is well written and easy to follow in its logic.

Materials and methods are described very clearly and with much detail.

Re: Thank you.

Major concern: The manuscript lacks relevant positive controls. Multiple monoclonal antibodies have been approved for the treatment of SARS-CoV-2 infection, either with the prototypic Wuhan strain or the currently circulating omicron variants. Including one of these antibodies (or any other well-characterized and published antibody, for example antibody VHH-72-Fc, described here: doi: 10.1038/s41467-020-19684-y. and available through ThermoFisher) would allow the reader to better understand how strong the neutralizing and antiviral activity of the two described IgYs is.

Re: We used our published human recombinant monoclonal antibody 23G7 (https://pubmed.ncbi.nlm.nih.gov/33080984/) as internal control in some of our experiments because the Ab was not able to neutralize emerging strains (unpublished results), and therefore it was excluded as a control from our study (unpublished confidential data).

Fig 1x. SARS-CoV-2 virus neutralizing assay in VERO E6 cells for determining IC50 curves for SARS-CoV-2 neutralizing IgG clones. The virus was pre-treated with serially diluted antibodies and infection of VERO E6 cells was determined based on cell viability 72 h post infection.

Fig 2x. Pseudovirus nautralization assay of 23G7 antibody with Wuhan, Alpha, Beta, Delta, Omicron and Omicron BA.2 VoC.

Minor Concerns:

  • The manuscript would benefit from a more substantial discussion. The role of Mabs for the treatment of SARS-CoV-2 should be mentioned. A comparison between the egg yolk IgYs described here and the clinically used Mabs in terms of antiviral activity, development timelines, regulatory aspects and scale-up/purification for large quantities might be of interest to the readers

Re: We have now described the role of mAbs in the manuscript: “Several recombinant monoclonal antibodies that bind to the viral trimeric S protein and prevent binding with the host receptor angiotensin-converting enzyme 2 (ACE2) have been approved are in clinical trials because they elicited preliminary clinical impact in limiting the development of severe COVID-19 when administered during early onset of infection [3-5]. However, intravenous administration, high concentration and high costs represent important limitations to the accessibility of these therapeutics and the need for additional more efficacious approaches remains.”

“By contrast to purified monoclonal antibodies, only 2-10% of polyclonal chicken IgYs are specific to their antigens [25-28], and, thus, higher concentration of IgYs is required to achieve therapeutic/prophylactic effect against SARS-CoV-2 in vitro and in vivo. There-fore, purification of antigen specific IgYs could be an important step to improve antiviral efficacy of IgYs.”

  • Through-out the manuscript: TCID50 (50 should be subscripted)

Re: Corrected.

  • Legend to Figure 1d/e; unclear which graph is Wuhan and omicron RBD

Re: Corrected: “(d) Binding of anti-RBD_Wuhan, anti-RBD_Omicron, or control IgYs to Wuhan RBD measured by indirect ELISA (OD at 650 nm). Mean; n = 4 (2 experiments with 2 parallels). (e) Binding of anti-RBD_Wuhan, anti-RBD_Omicron, or con-trol IgYs to Omicron RBD measured by indirect ELISA (OD – 650 nm). Mean; n = 4 (2 experiments with 2 parallels).”

  • A discussion of the extent of cross-reactivity of the Wuhan IgY to the Omicron RBD and vice versa would be interesting. Is this expected?

Re: It is expected. But further MOA studies (individual Ab purification and S-Ab structure determination) are needed to address this point.

  • In the hamster study, animals are dosed and inoculated intranasally. Please discuss the possibility if the IgYs might neutralize the viral inoculum in the nasal cavity.

Re: Thank you. We added your idea to the discussion: “Moreover, the IgYs might neutralize the viral inoculum in the nasal cavity.”

  • In the hamster study, a prophylactic dosing regimen was used. Have you considered a therapeutic dosing regimen?

Re:  Group A (6 animals) was treated with IgY raised against RBD of S protein of SARS-CoV-2/Wuhan (4,6 mg IgY/dose) intranasally 1 h before infection and then twice per day for 3 days.

  • What was the dose of the IgY antibody in group A? Were other doses explored? If not, why?

Re:  Group A (6 animals) was treated with IgY raised against RBD of S protein of SARS-CoV-2/Wuhan (4,6 mg IgY/dose) intranasally 1 h before infection and then twice per day for 3 days. We did not repeat the experiment, because others published the results of optimized study.

  • Figure 2: similar to panel C, the activity of the two IgYs against the Wuhan strain would be interesting

Re:  We agree. According to our data anti-RBD-Omicron IgYs are better than anti-RBD-Wuhan IgYs.

  • Figure 3b: statistical analysis in the figure (as done in panel c and d) rather than in the legend would be nice •

Re:  We agree. We show now the statistical analysis.

(b) Lesion scores in the lungs of golden Syrian hamsters. Statistical evaluation was perfromed using Wilcoxon rank-sum test (*** - p <0.001). Mean ± SD; n = 6.

  • Figure 3: instead of group A, B, C, legends with the treatment would be helpful (infected and treated; infected but untreated,…)

Re:  Done.

Round 2

Reviewer 2 Report (New Reviewer)

The authors provided a revised manuscript within 24 hours and addressed all minor concerns adequately. However, the major concern (lack of a positive control) has not been resolved. In their cover letter, the authors indicate, that a previously published antibody (23G7) has been included as a positive control in some experiments. Adding these results would address the issue. At the bare minimum, the discussion should include a comparison of the antiviral activity of the two IgYs (EC50 in mg/mL range) with 23G7 (EC50 in ng/mL) from their previous publication (https://pubmed.ncbi.nlm.nih.gov/33080984/).

Author Response

Many thanks for your comments. We added the results with 23G7 as positive control (please see Fig. S1 attached). We also discussed  the comparison of antiviral activity of IgYs (EC50 in mg/mL range) with 23G7 (EC50 in ng/mL range) in the text: "

By contrast to purified monoclonal antibodies, only 2-10% of polyclonal chicken IgYs are specific to their antigens [25-28]. Thus, by contrast to purified monoclonal antibodies, such as 23G7 (EC50 in ng/mL range) [21], 1000-fold higher concentrations of IgYs (EC50 in mg/mL range) are required to achieve anti-SARS-CoV-2 effect in vitro (Fig. S1). The same could be applicable for in vivo activity of IgYs vs. monoclonal antbodies. Therefore, purification of antigen specific IgYs could be an important step to improve antiviral efficacy of IgYs.

"

This manuscript is a resubmission of an earlier submission. The following is a list of the peer review reports and author responses from that submission.

Round 1

Reviewer 1 Report

The study addressed an important and inexpensive technology to prevent SARS-Co V2 infection. Although other studies reported similar results because the commercial value of the technology is limited, publishing more studies indicating the efficacy of this approach may trigger interest from a not-for-profit agency to take it on. I have the following small comments: 

1. Why refer to the technology as a Norwegian platform when publications from Turkey, Egypt, Australia, India, Canada, Peru, China, Finland, and Spain (to name a few) use the same technology?

2. The exact RBD amino acid range should be provided in the abstract and methods.

3. The discussion (2nd paragraph) claims "no toxicity'. However, the toxicity was not assessed in this study

4. Figure S1 should be included in the body of the study. It indicates an important value of the approach and should not be 'hidden' in the supplemental material

5. A simple literature search "IgY and SARS" provide ~20 publications, and at least some of these studies should be quoted.

6. Can the authors provide any histopathology in the lungs of the infected hamsters?

Author Response

Referee 1:

The study addressed an important and inexpensive technology to prevent SARS-CoV-2 infection. Although other studies reported similar results because the commercial value of the technology is limited, publishing more studies indicating the efficacy of this approach may trigger interest from a not-for-profit agency to take it on. I have the following small comments:

R1_1: Why refer to the technology as a Norwegian platform when publications from Turkey, Egypt, Australia, India, Canada, Peru, China, Finland, and Spain (to name a few) use the same technology?

Re: We agree. We modified the title accordingly: ‘Antiviral immunoglobulins of chicken egg yolk for prophylactics of SARS-CoV-2 infection’.

R1_2:  The exact RBD amino acid range should be provided in the abstract and methods.

Re: We have now provided the exact RBD amino acid in the methods: ‘RBD antigens of Wuhan and  Omicron SARS-CoV-2 variants (His-tagged RBDs of S proteins of SARS-CoV-2/Wuhan (319-541 aa) , The Native Antigen Company, REC31882, and SARS-CoV-2/Omicron (319-552 aa), Sino Biological, 40592-V08H121) were diluted in PBS to obtain 20 µl volume and mixed and emulsified with 180 µl Imject™ Freunds Complete adjuvant (Thermo Scientific)’.

R1_3: The discussion (2nd paragraph) claims "no toxicity'. However, the toxicity was not assessed in this study

Re: We agree. We removed the statement from the discussion. However, we done a similar study in hamster where we checked the appearance of IgY in blood after different applications, showing that IN applications gave non-detectable levels of IgY in blood which is in contrary to IP administration and thereby strongly suggest that IN administration is very safe. In addition, other study reported that intranasal anti-SARS-CoV-2 RBD IgY preparation had an excellent safety profile in rats following intranasal delivery of the formulated IgY. Moreover, the study showed no binding (cross-reactivity) to a variety of human tissues. In addition, A double-blind, randomized, placebo-controlled phase 1 study evaluating single-ascending and multiple doses of anti-SARS-CoV-2 RBD IgY administered intranasally in healthy adults also demonstrated an excellent safety and tolerability profile, and no evidence of systemic absorption (PMID: 35720389). We modified the discussion accordingly.

R1_4:  Figure S1 should be included in the body of the study. It indicates an important value of the approach and should not be 'hidden' in the supplemental material

Re: Figure S1 is now included as Fig. 2c in the body of the study: ‘. Importantly, IgYs raised against RBDs of S proteins of Wuhan and Omicron variants were also able to neutralize Sars-CoV-2/Omicron variant (Figure 2c).’

R1_5:  A simple literature search "IgY and SARS" provide ~20 publications, and at least some of these studies should be quoted.

Re: We have now quoted some of the studies in the discussion section of our manuscript: ‘Further studies will be required to determine if the IgY-based approach that was witnessed for in situ, in vitro and in vivo models and phase 1 clinical trial [19-22] could be translated in clinical phase 2 trials. If successful, IgY-based antiviral therapeutics could be valuable due to their lowered potential for adverse side effects (except for peo-ple with egg-allergies), and thus be useful in treating patients with emerging and re-emerging viral diseases. In addition, the IgY could be delivered through inhalation and other routes, leading to greater ease of treatment. Furthermore, IgY -based technology could be used in future viral outbreaks in Norway, given the demonstrated proof-of-concept.

Thus, our and other proof-of-concept pilot studyies shows that research on potential antiviral IgYs can have significant local as well as global impact, by increasing protection of the population against emerging and re-emerging viral diseases and filling the time between virus identification and vaccine development with life-saving countermeasures [23,24].’

R1_6:  Can the authors provide any histopathology in the lungs of the infected hamsters?

Re: In our hamster experiment, in which IgYs were administered IN, the lung samples were collected on D4 and histomorphometric changes where scored, we did not detect significant differences in inflammation and hemorrhage markers between groups. However, in our other study (will be published separately) when IgY preparation was administered IP, we observed very good response in histopathology. It seems that IgY administrated IN by contrast to IP does not reach the lungs. This agrees with published research {Wongso, 2022 #104} stating that when IgY is administered IN most IgY accumulates in the trachea in mice.

Reviewer 2 Report

The authors of the manuscript entitled “Norwegian platform for production of antiviral chicken egg yolk immunoglobulins for viral pandemics preparedness” have investigated the establishment of a protocol for the generation of IgY antibodies against the RBD of SARS-CoV-2 as a model for the preparedness for the COVID pandemic and future potential pandemic infections. The idea is very promising given the previous reports on the potential use of IgY against viral infections. 

The manuscript is well written, and the introduction, methods and results are clearly written. The following comments are needed to increase the clarity of the study:

Major comments

The authors have demonstrated the binding of the IgY to the target RBD protein using ELISA and compared it to lack of binding of the control IgY (from non-immunized hens), however the specificity of this binding should have been better demonstrated by Western Blot 

Minor Comments:

1.     In the section “Anti-RBD IgY Testing and Data Quantification”, It is not clear whether the cells were infected after or before the addition of the IgY. The authors need to clarify the time of addition of IgY to the virus.

2.     Assays for omicron variants neutralization is not mentioned in the methods section.

Author Response

Referee 2:

Major comments

R2_1: The authors have demonstrated the binding of the IgY to the target RBD protein using ELISA and compared it to lack of binding of the control IgY (from non-immunized hens), however the specificity of this binding should have been better demonstrated by Western Blot

Re: We have performed SDS-PAGE and immunoblot analysis. Fig. 1b and c demonstrate the binding of the IgYs to the RBDs.

Minor Comments:

R2_2:    In the section “Anti-RBD IgY Testing and Data Quantification”, It is not clear whether the cells were infected after or before the addition of the IgY. The authors need to clarify the time of addition of IgY to the virus.

Re: We have now clarified the method section: ‘The medium was then replaced with DMEM containing 0.2% BSA, Pen–Strep, and the IgY preparation in 3-fold dilutions at 7 different concentrations. No IgY preparation was added to the control wells. After 15 min the cells were infected with virus at an moi of 0.01 or mock.’

R2_3:     Assays for omicron variants neutralization is not mentioned in the methods section.

Re: We have now mentioned the neutralization assays for omicron variant in the methods section: ‘After 48 h of infection, a CellTiter-Glo (CTG, Promega, Madison, WI, USA) assays were performed to measure viability of SARS-CoV-2-mCherry, of SARS-CoV-2/Omicron and mock-infected cells as described previously [11-13].’

Reviewer 3 Report

The manuscript reports the development of chicken polyclonal IgY against RBD of SARS-CoV-2. The anti-RBD IgY showed potent neuralization activity in vitro and lower viral RNA expression in the lungs of Syrian Golden Hamster.

Major issues:

1.     The author used a crude extract of IgY for the in vitro and in vivo studies. Chromatography purification and SDS-PAGE characterization of the IgY should be performed. The purified IgY should be used in the binding assays as well as animal study.

2.     Cell infection study: Author used a mCherry fluorescence assay to show the antiviral effect of the IgY against SARS-CoV-2. The data should be supported by in situ localization of the SARS-CoV-2 proteins in IFA and/or ISH assays.

3.     Hamster experiment:

a.     Author mentioned in the M&M that the animal viability, behavior and clinical parameters were monitored daily, however, no such data was provided in the results. Comparison of clinical parameters such as body weight, temperature, clinical signs between the three groups should be presented.

b.     Author showed only the viral RNA load in the lungs of the animals. What about other tissues and organs? Author should quantify and present the infectious virus load in swabs (oropharyngeal, nasal and cloacal) and tissues. It is important to see if the crude IgY reduces infectious virus load in tissues and lowers the shedding.

c.     A comparative picture of the gross and histopathological changes in tissues, in particular the lungs should be presented between groups.

4.     The discussion section requires significant improvement. Data should be discussed in terms of recent literature. The manuscript contains a surprising number of self-citations.

Other issues:

1.     Title: The title is very broad and does not reflect the content of the study. 

2.     What is the lineage of the challenge virus? The titer of the challenged virus is not clear, pfu/TCID50, which unit was used? 

3.     Quantification of viral RNA should be presented as copy number or as Ct. Author could use a standard curve to quantify the copy number of virus in the tissues/swabs per unit of samples.

4.     Figure 1: Author should use different symbols (for dot plot) or patterns (for column) while showing different groups.

Author Response

Referee 3:

Major issues:

R3_1: The author used a crude extract of IgY for the in vitro and in vivo studies. Chromatography purification and SDS-PAGE characterization of the IgY should be performed. The purified IgY should be used in the binding assays as well as animal study.

Re: We have performed SDS-PAGE analysis of our IgY preps (Fig. 1b). It should be noted that crude IgY preps are used in such studies to show the potential of antiviral IgYs as a prophylactic option.

R3_2: Cell infection study: Author used a mCherry fluorescence assay to show the antiviral effect of the IgY against SARS-CoV-2. The data should be supported by in situ localization of the SARS-CoV-2 proteins in IFA and/or ISH assays.

Re: Previous studies showed that anti-SARS-CoV-2 RBD IgY preparation had no binding/cross-reactivity to a variety of human tissues [19]. Therefore, IFA and/or ISH assays could be inappropriate to support our in vitro results.

R3_3: Hamster experiment:

  1. Author mentioned in the M&M that the animal viability, behavior and clinical parameters were monitored daily, however, no such data was provided in the results. Comparison of clinical parameters such as body weight, temperature, clinical signs between the three groups should be presented.

Re: Indeed, we monitored animal viability, behavior and clinical parameters. However, we did not detect significant differences between the groups. We have now mention this in the manuscript.

  1. Author showed only the viral RNA load in the lungs of the animals. What about other tissues and organs? Author should quantify and present the infectious virus load in swabs (oropharyngeal, nasal and cloacal) and tissues. It is important to see if the crude IgY reduces infectious virus load in tissues and lowers the shedding.

Re: Other organs than the lungs were not included in the study. Further experiments are needed to address this and viral load issues, however they are cost and time-consuming and ae against 3R policy.

  1. A comparative picture of the gross and histopathological changes in tissues, in particular the lungs should be presented between groups.

Re: In our hamster experiment, in which IgYs were administered IN, the lung samples were collected on D4 and histomorphometric changes where scored, we did not detect significant differences in inflammation and hemorrhage markers between groups. However, in our other study (will be published separately) when IgY preparation was administered IP, we observed very good response in histopathology. It seems that IgY administrated IN by contrast to IP does not reach the lungs. This agrees with published research {Wongso, 2022 #104} stating that when IgY is administered IN most IgY accumulates in the trachea in mice.

R3_4:     The discussion section requires significant improvement. Data should be discussed in terms of recent literature. The manuscript contains a surprising number of self-citations.

Re: We have now substantially improved the discussion section and added citations to recent literature.

Other issues:

R3_5:          Title: The title is very broad and does not reflect the content of the study.

Re: We agree. We modified the title: ‘Antiviral immunoglobulins of chicken egg yolk for prophylactics of SARS-CoV-2 infection’.

R3_6:         What is the lineage of the challenge virus? The titer of the challenged virus is not clear, pfu/TCID50, which unit was used?

Re: Slovakia/SK-BMC5/2020 (B.1.1 lineage) was isolated from a COVID-19 patient from Slovakia in March 2020. Its complete sequence was deposited on GISAID.org under the accesion ID EPI_ISL_417879.

R3_7:   Quantification of viral RNA should be presented as copy number or as Ct. Author could use a standard curve to quantify the copy number of virus in the tissues/swabs per unit of samples.

Re: The parameter is mean 2-D CT (the house keeping gene ORF1a is compared to b-actin).

R3_5: Figure 1: Author should use different symbols (for dot plot) or patterns (for column) while showing different groups.

Re: We now made a separate figure for the in vivo results to avoid confusions (Fig. 3).

Round 2

Reviewer 3 Report

1. The author partially addressed the comments posted in the previous round of revision. My major concern would still be in the animal study. Hamster is a very good model for SARS-CoV-2. Many studies showed that hamsters inoculated with SARS-CoV-2 show significant body weight loss, severe pneumonia, efficient virus replication in the nasal turbinate, trachea and lungs with marked inflammatory responses.

2. Author should provide sufficient evidence that the control untreated hamsters show signs and pathology of productive SARS-CoV-2 infection. After that, one can appreciate if the IgY treatment prevented/alleviated the SARS-CoV-2 infection in the treated group. Author mentioned that there was no differences in the clinical parameters between treated and untreated groups. Did the infected hamsters show body weight loss?

3. Moreover, no data on lung pathology were presented or even mentioned in the result or discussion. Although, the author mentioned in the response to the reviewer section that they did not see any differences between groups. Please mention clearly in the results: did the animals show lung pathology? Were there any differences in lung inflammation between treated, untreated and control animals? Did the virus replicate in the lungs (IHC/ISH)?

4. The author claimed that the IgY preparation neutralized SARS-CoV-2 in vivo. Unfortunately, based on the data provided, it is hard to justify the statement. The only animal experimentation result the author provided was the qPCR data showing the viral RNA and some cytokine mRNA in the lungs. As the animals were inoculated via the intranasal route with the SARS-CoV-2, detection of viral RNA in the lungs is expected and can't be used as a proof of successful infection in the animal. It is important for the study to show the comparative data of clinical illness (body weight loss), lungs pathology and infectious virus in the lungs (TCID50 and ISH), in both control untreated and IgY immunized hamsters, so that readers could appreciate the importance of IgY as an effective treatment option.

5. To me it is not clear why the author chose to sample the lungs only. SARS-CoV-2 is known to replicate in nasal turbinate, trachea and lungs and produce inflammatory changes in these tissues. 

Author Response

Referee 3:

  1. The author partially addressed the comments posted in the previous round of revision. My major concern would still be in the animal study. Hamster is a very good model for SARS-CoV-2. Many studies showed that hamsters inoculated with SARS-CoV-2 show significant body weight loss, severe pneumonia, efficient virus replication in the nasal turbinate, trachea and lungs with marked inflammatory responses.
  2. Author should provide sufficient evidence that the control untreated hamsters show signs and pathology of productive SARS-CoV-2 infection. After that, one can appreciate if the IgY treatment prevented/alleviated the SARS-CoV-2 infection in the treated group. Author mentioned that there was no differences in the clinical parameters between treated and untreated groups. Did the infected hamsters show body weight loss?
  3. Moreover, no data on lung pathology were presented or even mentioned in the result or discussion. Although, the author mentioned in the response to the reviewer section that they did not see any differences between groups. Please mention clearly in the results: did the animals show lung pathology? Were there any differences in lung inflammation between treated, untreated and control animals? Did the virus replicate in the lungs (IHC/ISH)?
  4. The author claimed that the IgY preparation neutralized SARS-CoV-2 in vivo. Unfortunately, based on the data provided, it is hard to justify the statement. The only animal experimentation result the author provided was the qPCR data showing the viral RNA and some cytokine mRNA in the lungs. As the animals were inoculated via the intranasal route with the SARS-CoV-2, detection of viral RNA in the lungs is expected and can't be used as a proof of successful infection in the animal. It is important for the study to show the comparative data of clinical illness (body weight loss), lungs pathology and infectious virus in the lungs (TCID50 and ISH), in both control untreated and IgY immunized hamsters, so that readers could appreciate the importance of IgY as an effective treatment option.
  5. To me it is not clear why the author chose to sample the lungs only. SARS-CoV-2 is known to replicate in nasal turbinate, trachea and lungs and produce inflammatory changes in these tissues. 

Re: We agree with the reviewer. Indeed, our in vivo results are weak due to study design limitations. We have now added the detailed protocol and study results obtained by CRO as supplementary file. We also present and discuss the results and limitations in the main body of the manuscript:

Due to study design limitations, we were unable to demonstrate viral neutralization (Supplementary file). Moreover, we did not detect significant differences in viability, behavior, and clinical parameters between the non- and IgY-treated groups of hamsters challenged with SARS-CoV-2 (Supplementary file).’

Anti-RBD IgYs lowered viral RNA amplification in lungs of in hamsters challenged with SARS-CoV-2. At the same time the treatment allowed activation of several innate immune-related genes, that mediate the development of adaptive immune responses. Nevertheless, we were unable to demonstrate viral neutralization (Supplementary file). This may be associated with using too much virus in this hamster model of COVID-19 (105 TCID50 in our study vs. 1 x 104 or 5 x 104 TCID50 in published studies [20] (https://doi.org/10.1101/2020.10.28.359836). This agrees with previous study [19]. We as well as the authors of that study also noted that IN formulated IgY preparation was viscous. As hamsters are obligatory nose-breathers, they may have blown out the formulated IgY. Finally, the virus that was delivered in 50 μL of liquid directly into each nare may have washed out some antibodies. However, a recent study demonstrated a protective effect of IN administration of anti-RBD IgY in hamsters challenged with SARS-CoV-2 [21], which suggests that our prophylactic if optimized could also be protective.’

Recent study demonstrated a protective effect of IN administration of anti-RBD IgY in hamsters challenged with SARS-CoV-2 [21], which suggests that our prophylactic treatment if optimized could also be protective. However, we do not have resources and capacity (3R policy) to perform such study

If the reviewer considers that the in vivo section is weak, we may remove it from our manuscript.